# Applications of Genomics and Transcriptomics in Precision Medicine for Myopia Control or Prevention

**DOI:** 10.3390/biom13030494

**Published:** 2023-03-07

**Authors:** Liqin Jiang, Dong Xuan Goh, James Hao Zhong Koh, Xavier Chan, Noel A. Brennan, Veluchamy Amutha Barathi, Quan V. Hoang

**Affiliations:** 1Singapore Eye Research Institute, Singapore National Eye Centre, Eye ACP Duke-NUS Medical School, Singapore 169856, Singapore; 2Lee Kong Chian School of Medicine, Nanyang Technological University, Singapore 636921, Singapore; 3Johnson & Johnson Vision, Jacksonville, FL 32256, USA; 4Department of Ophthalmology, Yong Loo Lin School of Medicine, National University of Singapore, Singapore 119228, Singapore; 5Department of Ophthalmology, Columbia University, New York, NY 10032, USA

**Keywords:** Myopia, molecular, genomics, GWAS, transcriptomics, animal models

## Abstract

Myopia is a globally emerging concern accompanied by multiple medical and socio-economic burdens with no well-established causal treatment to control thus far. The study of the genomics and transcriptomics of myopia treatment is crucial to delineate disease pathways and provide valuable insights for the design of precise and effective therapeutics. A strong understanding of altered biochemical pathways and underlying pathogenesis leading to myopia may facilitate early diagnosis and treatment of myopia, ultimately leading to the development of more effective preventive and therapeutic measures. In this review, we summarize current data about the genomics and transcriptomics of myopia in human and animal models. We also discuss the potential applicability of these findings to precision medicine for myopia treatment.

## 1. Introduction

Myopia persists as a pressing public health issue globally, with 27% of the population (or 1.45 billion) involved in 2010 and numbers foreseen to rise [1]. The prevalence of myopia among schoolchildren of Asian descent has been rising significantly from 5.8% in 1983 to 21% in 2000 [2]. The rising prevalence of myopia was also reflected in a study of European adults [3].

Quantitatively, human myopia is commonly defined as having a spherical equivalent (SE) refraction of ≤−0.50 diopter (D), and categorized as mild myopia when SE ≤ −3.00 D and moderate myopia between −3.00 D and >−6.00 D. Once SE ≤ −6.00 D, it is categorized as high myopia (HM), in which the axial length (AL) is usually >26 mm [4]. Myopic children usually become more nearsighted as they age, but their refractive error usually stabilizes in their 20s.

Those with HM will be at higher risk of developing pathological complications such as macular degeneration, choroidal neovascularization, chorioretinal atrophy, and retinoschisis which may result in vision deterioration, ultimately leading to blindness, especially in patients that are 60 years old or over [5].

Myopia that is not associated with another disease is known as non-syndromic myopia, while patients with syndromic myopia are predisposed to myopia due to a disease that is usually genetically inherited, which makes a clinical distinction important. Flitcroft and colleagues have reported that variants of genes known to cause syndromic myopia are also present in non-syndromic myopia [6].

Key contributors to myopia may be attributed to the interactions between environmental and genetic factors. Environmental factors such as near-work activities are reviewed and researched again by Hepsen and colleagues to be associated with the onset and progression of myopia [7], and Huang and colleagues concluded a 2% increased chance of developing myopia with every diopter-hour of near work time per week [8]. Genetic factors are also involved in the development and progression of myopia, exhibited by a study showing Asian children being more susceptible to high myopia as compared to their European counterparts by analysis of 20 different myopia-associated loci [9]. The genetic architecture of myopia and currently associated myopia loci, as well as gene-environment interactions and the prediction of myopia via polygenic risk scores have been comprehensively examined previously [10]. Other factors include high educational pressure and limited time outdoors [11]. Limiting educational workloads and increasing time outdoors have been suggested to improve outcomes in the myopic progression of pediatric patients.

Understanding the various factors underpinning myopia may provide valuable insights for the development of therapeutic options for myopia. Therefore, animal myopia models from different species have been proposed to study myopia, each with their pros and cons. Lens-induced myopia (LIM) or form deprivation myopia (FDM) are the two primary ways to induce detectable myopia for better control of environmental factors, and genetically-manipulated animal models, mainly mice, are preferred to study syndromic myopia in humans [12].

Limited treatment options currently available to retard axial elongation in myopia highlights the imperative need for work to be conducted in the field of myopia pharmacogenetics [13,14]. Myopia pharmacogenetics can be used synergistically with precision medicine, which detects at-risk populations, such as the early onset of myopia during childhood, and pathological myopia development during adulthood. Precision medicine has been applied in other fields of ophthalmology, such as in the management of Age-Related Macular Degeneration [15], and has only very recently been applied in the management of pediatric myopia [16].

Molecular biology has been comprehensively reviewed in myopia research to provide perspectives on the genetic basis of ocular development and probable pharmacological candidates for intervention [17]. Many tremendous and exciting advances are taking place within the biological field, facilitated by numerous biotechnological developments and applications [18]. Such advancement has led to the discovery of novel genes that have been implicated in the development and progression of myopia [19], such as in dopamine signaling [20].

## 2. Significance of Genetic Analyses

The central dogma of molecular biology states that biological information is encoded by DNA and the genetic information is transcribed to messenger RNA. Messenger RNA is then translated into proteins [21]. However, other factors can influence the production of proteins. Epigenetics investigates the changes in gene expression associated with alterations in the chromosome, while the DNA sequence remains unchanged. This gives more information regarding the interaction of genetic and environmental factors [22]. Transcriptomics, on the other hand, involves analyzing which of the genes encoded in DNA are turned on or off and to what extent. Transcriptomics is a continuum that complements genomics, filling the gap between genomics and proteomics in the precision medicine era. The two key contemporary techniques employed in transcriptomics include RNA-sequencing (RNA-seq) to examine the quantity and sequences of RNA in a sample using next-generation sequencing (NGS), and microarray, which detects thousands of genetic sequences simultaneously [23].

## 3. Techniques of Genomics and Transcriptomics

### 3.1. Genomics

The term ‘genomics’ was coined to define the study of genes and their functions [24,25]. This has given rise to the revolutionary Human Genome Project in 1990, which cataloged the genetic sequences of the human body [26]. The genome includes chromosomal and extrachromosomal DNA material, consisting of both coding and non-coding regions.

Linkage analysis, previously the main, powerful analysis method for the identification of genes involved in disease etiology, can be conducted in conjunction with NGS or whole exome sequencing approach [27].

At present, genome-wide association study (GWAS) is the main research approach used to identify genomic variants (such as single nucleotide polymorphism [SNP], a genomic variant at a single base position in the DNA) that are statistically associated with a risk for a disease or a particular trait. GWAS results can be applied in many settings, such as helping to gain insights into a phenotype’s underlying biology, calculating genetic correlations, making clinical risk predictions, and informing drug development programs about risk factors and health outcomes. To validate GWAS results, a meta-analysis is usually used to reduce the chance of false positives; quantitatively, different populations from similar ancestry are used to run GWAS again to confirm the signals. Other methods include targeted re-sequencing of the regions of interest, custom SNP Chip analysis, and microarray.

In terms of bench work for validation, Polymerase Chain Reaction (PCR) is used to amplify small segments of DNA to attain large quantities of DNA [28]. It is the most common DNA amplification method used and it facilitates the analysis of genes involved in myopic development and progression. PCR is performed with a thermostable DNA polymerase [29]. Southern blotting involves covalently binding DNA fragments to membranous support, which can then be used for hybridization [30]. This helps in the detection of specific DNA sequences in complex samples of DNA, which can be applied in the context of myopia genomics. Northern blotting is similar to Southern blotting, but it involves the detection of RNA sequences instead of DNA sequences and was therefore used heavily in the field of transcriptomics [31].

Microarray greatly advances the study of differential genetic expressions by allowing for the analysis of thousands of genetic sequences [32]. A probe is used to detect the presence of complementary genetic sequences and therefore genetic variants.

### 3.2. Transcriptomics

RNA-seq is a sophisticated molecular technique that sheds light on gene expression in cells in physiological and pathological states [33]. This aids in the understanding of the molecular mechanisms underlying myopic refractive status. For example, RNA-seq has led to the discovery that the retina is likely able to distinguish between hyperopia and myopia [34].

## 4. Molecular Technology Revolution

The usage of molecular technology, along with animal myopia models, provides vast amounts of information and aids in providing an overview of biological data. Identifying specific signaling pathways underlying myopia could allow the discovery of potential treatments for myopia [35].

Molecular technology has led to the discovery of several specific pathways underlying the pathogenesis of myopia. These pathways are likely to involve cross-signaling between the layers of the retinal fundus. For instance, the thickening of the choroid has been associated with a decrease in the synthesis of extracellular matrix molecules including glycosaminoglycan (GAG) chains. The amount of GAG chains produced by the choroid might serve as a reliable indicator for the rate of eye growth [36], suggesting the involvement of the choroid in directing scleral growth.

With a wide array of techniques to assay genes, such as single-cell sequencing and RNA-seq, along with the increased sensitivity and specificity of test assays, integrative analyses of both genomics and transcriptomics could provide promising data for illustrating the genetic architecture underpinning normal eye growth and myopia development, which is a key step to make a more plausible hypothesis of the pathways underlying myopia.

Recent investigational methodologies involve the analysis of pathways involving the utilization of genomics and transcriptomics data. Pathway analysis uses bioinformatic data obtained from high-throughput technologies such as RNA-seq, in order to identify relevant groups of related genes that are altered in diseased samples compared to a control [37]. Currently, the most popular method of pathway analysis is the Gene Set Enrichment Analysis (GSEA), which references pathway databases such as Gene Ontology (GO) and Kyoto Encyclopedia of Genes and Genomes (KEGG) in order to output relevant molecular pathways represented in the form of lists, plots, links to interactive pathway diagrams, and/or network visualizations. Software, such as Ingenuity Pathway Analysis (IPA, QIAGEN), has been made commercially available for this purpose (Available online: https://digitalinsights.qiagen.com/products-overview/discovery-insights-portfolio/analysis-and-visualization/qiagen-ipa/ (accessed on 9 January 2023)).

## 5. Similarity of Eye Structure and Cell Types between Species

The basic retinal structure of vertebrates is closely related and is usually characterized by the presence of five major layers and five major cell types [38]. This enables the translation from animal work to human studies. For example, rod cells have been found to be central to the pathogenesis of FDM in *Gnat1* gene knockout mice [39], while in myopia patients, alterations in photoreceptor density were found using an ocular imaging technique as reduced rod and cone ratio [40], and reduced astrocyte count [41], while functional electroretinogram (ERG) recording indicated an association of the rod system function with the progression of myopia [42]. Additionally, SNP rs6979985 of the *VIPR2* gene was found to be associated with high myopia in a Chinese Han cohort. Animal studies demonstrated a refractory shift towards myopia for the cohort of *Vipr2* knockout mice. Moreover, ERG recording showed a compromised bipolar cell function in the retina of *Vipr2* knockout mice [43].

Other animal studies demonstrated that myopic eye growth is a locally controlled disease [44,45], where a gene mutation affecting a local structure would manifest as a refractive error. Moreover, in humans, mutations in the gene encoding the major component collagen type II (*COL2A1*) in the sclera have been associated with high congenital myopia [46]. Furthermore, an autosomal dominant disorder due to mutations in the fibrillin gene encoding elastic fibers in the sclera is evidenced by the scleral pathology and high myopia associated with Marfan syndrome [47]. Therefore, findings from animal models are a very valuable resource in research with findings readily translatable to humans.

## 6. Applications of New Technologies in Myopia Research

Big data analytics can reveal correlations in large amounts of raw genetic data and calculates the risk contributed by environmental factors leading to the development of myopia in order to help clinicians make data-informed and clinically significant decisions. Myopia research has been applied successfully in many cases. The first successful applications were in GWAS studies reported by Nakanishi and colleagues in an Asian population, as well as Solouki and colleagues in a European population [48,49]. More recent studies listed in Table 1 highlight the successful application of myopia research in human studies, from identifying more SNPs to specific genes in different races. Studies also identified specific genes that are heavily influenced by the environment, with education being a salient contributing factor. GWAS studies in the human population emphasized the role of the retina in the generation of myopic eye growth signaling and supported the notion that refractive errors are caused by a light-dependent retina-to-sclera signaling cascade.

Using genome-wide survival analysis, Kiefer and colleagues performed the GWAS meta-analysis from the 23andMe database and identified 22 significant associations with the age of onset of myopia with 45,771 European participants. The associations in total explained 2.9% of the variance in myopia age of onset and pointed toward multiple genetic factors involved in the development of myopia. The associations also suggested that complex interactions between extracellular matrix remodeling, neuronal development, and visual signals from the retina may underlie the development of myopia in humans [50]. A more recent study suggests that mutations in the cone opsin genes were shown to account for about 4.6% of the variance in common myopia [51]. Following studies contributed by Tedja et al. combined the Consortium for Refractive Error and Myopia (CREAM) and 23andMe, expanding the study sample to 160,420 individuals from a mixed ancestry population, in addition to independent replication in 95,505 participants from the UK BioBank. System comparisons were conducted and showed a high genetic correlation between Europeans and Asians (>0.78, *p* = 2.48 × 10^−7^). Expression experiments and comprehensive in silico analyses identified retinal cell physiology and light processing as prominent mechanisms and identified functional contributions to refractive-error development in all cell types of the neurosensory retina, RPE, vascular endothelium, and extracellular matrix. Newly identified genes implicate novel mechanisms such as rod-and-cone bipolar synaptic neurotransmission, anterior-segment morphology, and angiogenesis. Thirty-one loci resided in or near regions transcribing small RNAs, thus suggesting a role for post-transcriptional regulation [52]. All these would be good bases for hypotheses to design animal studies for further validation and pharmaceutical target identification.

**Table 1 biomolecules-13-00494-t001:** GWAS in humans indicated that refractive error is influenced by both common and rare variants with a significant environmental component.

Publications	Research Techniques	Cohort Sizes	Findings
PMID: 35031440 Simcoe et al. (2022) [53]	Three-stage GWAS: discovery with three cohorts, replication with one cohort, meta-analysis with all four cohorts	574 cases with pigment dispersion syndrome (PDS) or pigmentary glaucoma (PG) and 52,627 controls of European descent	*GSAP* and *GRM5*/*TYR* genes are associated risk factors for the development of PDS, PG, and myopia.
PMID: 35841873 Xue et al. (2022) [54]	Cross-trait meta-analysis in age-related macular degeneration (AMD), diabetic retinopathy (DR), glaucoma, retinal detachment (RD), and myopia	43,877 cases of five ocular diseases: AMD, DR, glaucoma, RD, and myopia, and 44,373 controls of European ancestry	Three pleiotropic loci and genes covered positively associated with all five diseases: rs7678123 (*FGF5*, *C4orf22*, *BMP3*, *PRKG2*), rs12570944 (*STK32C*, *LRRC27*, *PWWP2B*, *DPYSL4*), and rs9667489 (*ME3*, *CCDC81*, and *PRSS23*).
PMID: 32231278 Hysi et al. (2020) [55]	Meta-analysis of GWASes from four cohorts. Mixed linear regressions, adjusting for age, sex, and the first 10 principal components	542,934 European participants	Identified 336 novel genetic loci associated with refractive error, driven by genes participating in the development of every anatomical component of the eye, as well as in circadian rhythm and pigmentation.
PMID: 25233373 Simpson et al. (2014) [56]	Meta-analysis of GWASes of myopia and hyperopia from nine cohorts	16,830 individuals of European ancestry	One genome-wide significant region on 8q12 was associated with myopia age at onset [50] and associated to mean spherical-equivalent (MSE) refractive error [57]. Found replication of 10 additional loci associated with myopia as reported by Kiefer et al. [50].
PMID: 23933737 Khor et al. (2013) [58]	Meta-analysis of four GWASes	East Asian descent totaling 1603 cases and 3427 controls	rs13382811 and rs6469937 are associated with severe myopia.
PMID: 27020472 Fan et al. (2016) [59]	Joint meta-analysis to test SNP main effects and SNP × education interaction effects on refractive error	40,036 adults from 25 studies of European ancestry and 10,315 adults from nine studies of Asian ancestry.	In European ancestry individuals, six novel loci were identified: *FAM150B-ACP1*, *LINC00340*, *FBN1*, *DIS3L-MAP2K1*, *ARID2-SNAT1*, and *SLC14A2*; In Asian populations, three genome-wide significant loci *AREG*, *GABRR1*, and *PDE10A* also exhibit strong interactions with education.
PMID: 24014484 Fan et al. (2014) [60]	Meta-analysis of the effects of education on 40 single nucleotide polymorphisms	8461 adults from five studies including ethnic Chinese, Malay, and Indian residents of Singapore.	Three genetic loci *SHISA6-DNAH9*, *GJD2*, and *ZMAT4-SFRP1* exhibited a strong association with myopic refractive error in individuals with higher secondary or university education.
PMID: 23406873 Shi et al. (2013) [61]	GWAS to examine the associations between myopia and 286,031 SNPs, with additional replication and 2 validation cohorts	Discovery: Han Chinese cohort of 665 cases and 960 controls, replication: 850 cases and 1197 controls, validation: combined 1278 cases and 2486 controls	The most significant and validated SNPs are located in *VIPR2* and *SNTB1*, which are expressed in the retina and RPE

Targeted gene studies can be achieved by classic molecular techniques utilized in studies listed in Table 2. Breakthroughs in the field of myopia research involve the application of molecular techniques and experimental animal models. For example, the use of PCR and Southern blotting has led to the discovery that the chick does not possess a functional M1 muscarinic acetylcholine receptor [62]. Such findings imply that muscarinic antagonists, which prevent the progression of myopia in the chick, either work through another muscarinic receptor subtype, most likely M4, or through non-specific or non-receptor mechanisms.

Another successful application is a study conducting drug screening for myopia control [63]. Researchers modified the lumican gene with a morpholino oligomer in zebrafish embryos, allowing the establishment of an excessive sclera expansion animal model for screening 642 compounds approved by the U.S. FDA. Effective compounds were then applied in form deprivation myopia models mice and the Syrian hamster. The study reported that MMP inhibitors are potential candidates for the treatment of myopia by targeting the sclera.

In addition to pharmacological interventions, effective optical interventions, informed by genetics, are already available [64]. Rappon et al. reported that the discovery of mutations in the cone opsin gene *OPN1LW* led to Bornholm Eye Disease (BED) as reported by McClements and colleagues [65]. Hagen and colleagues also reported that polymorphisms of the *OPN1LW* gene may play in role in non-syndromic common myopia [66]. The characterized biological function of the mutant genes [67] led to the development of diffusion optics technology (DOT) lenses (designed to reduce retinal contrast), which were shown to slow myopia progression by 74% in a multicenter, randomized, and controlled double-masked trial [16].

**Table 2 biomolecules-13-00494-t002:** Targeted genes in human myopia studies.

Publications	Research Techniques	Tissue Samples	Findings
PMID: 23322568 McClements et al. (2013) [68]	Long-range PCR; Quantitative PCR	Blood	BED is caused by a rare exon three haplotypes of *OPN1LW*.
PMID: 15197065 Young et al. (2004) [69]	PCR; Haplotype analysis of X chromosome; Southern Blot	Lymphocyte	X-linked high myopia associated with cone dysfunction may involve a defect in chromosome Xq28.
PMID: 31254532 Hagen et al. (2019) [70]	ERG; PCR; SNP genotyping	Saliva	The L:M cone ratio, combined with milder versions of *OPN1LW* polymorphisms, may play a role in non-syndromic myopia.
PMID: 33833231 Zhu et al. (2021) [71]	Gene expression microarray; qPCR; ChIP-qPCR;	Lens	The dysregulation of the *MAF-TGF-β1-crystallin* axis may be an underlying mechanism that leads to the development of high myopia.

Leveraging on harvested eye tissue from animal myopia models, differentially-expressed single genes analyzed from RNA-seq, along with right bioinformatic analysis methods, studies could elucidate the underlying mechanism of light-dependent signaling in the retina for myopic eye growth. Utilizing a short myopia induction period of 1–3 days in the chick model, Riddell N. et al. used GSEA and reported subtle shifts in structural, metabolic, and immune pathway expression in the retina, which were correlated with eye size and refractive changes induced by lens defocus [72]. Another study also utilized New World primates, common marmosets, which were treated with either negative (-5D) or positive (+5D) lenses for either a short period (10 days) or a long time (5 weeks) to induce refractive error. QIAGEN’s IPA software and database were used to identify biological functions (GO categories), which are significantly affected by the changes in gene expression induced by the optical defocus in the retina. IPA revealed that the primate retina responds to defocus of different signs by activation or suppression of largely distinct pathways. Twenty-nine genes that were differentially expressed in the marmoset retina in response to imposed defocus are localized within human myopia quantitative trait loci (QTLs), suggesting functional overlap between genes differentially expressed in the marmoset retina upon exposure to optical defocus and genes causing myopia in humans [34]. More recent transcriptomic work focused on the retina are detailed in Table 3 and Table 4.

## 7. Limitations

Various limitations of genomic and transcriptomic studies involving screening of pharmaceutical targets for myopia treatments are attributed to post-translational modifications, which contribute to the complexity from genome to proteome and regulates cell function [96]. The importance of genes related to myopia are also difficult to determine without also considering the protein expression levels of these genes. For example, Landis and colleagues have reported that although mRNA levels of tyrosine hydroxylase (*TH*) were significantly different in eyes under different light settings, there were no significant differences in the protein levels of *TH* [87]. Additionally, myopia is a polygenic disease [97], and numerous genes must be considered in conjunction when analyzing the pathogenesis of myopia. Therefore, multiple targets should be explored together to develop treatment options for this potentially blinding disease.

Furthermore, findings from some animal models may not be readily translatable to humans, therefore, discoveries from animal studies must always be considered carefully and in the proper context. For instance, the chick does not possess an M1 receptor, which is present in humans [62]. However, atropine, an M1 receptor antagonist, can inhibit myopia development in both chick and humans. As such, future studies could focus on fully understanding the molecular mechanisms of action behind atropine and its interaction with its receptor before we are able to yield data to substantiate the development of therapeutic drugs for the clinical treatment of myopia. In other species used as animal models, there are also several disadvantages. For instance, the mouse, tree shrew, and guinea pig models lack a fovea and also likely lack the ability for eye accommodation [98].

## 8. Conclusions

The general pipeline of precision medicine from genomic discovery to clinical translation can be found in Figure 1. As unprecedented volumes of sequence databases are being generated, coupled with new analytical platforms and integrated bioinformatics analysis, understanding data in the relevant scientific and medical context is vital in making big data interoperable and interpretable [99]. Myopia, a worldwide epidemic disease that is potentially blinding, can benefit from such data. Recent findings and breakthroughs, from pharmacogenomic approaches to develop drugs against myopia, to novel approaches proposed for clinicians to adopt in the clinical context, are promising [100]. Genetic studies would enable clinician and scientists to better control environmental factors to delay myopia onset or allow for myopia to be approached in new and innovative ways to slow down its development.

## Figures and Tables

**Figure 1 biomolecules-13-00494-f001:**
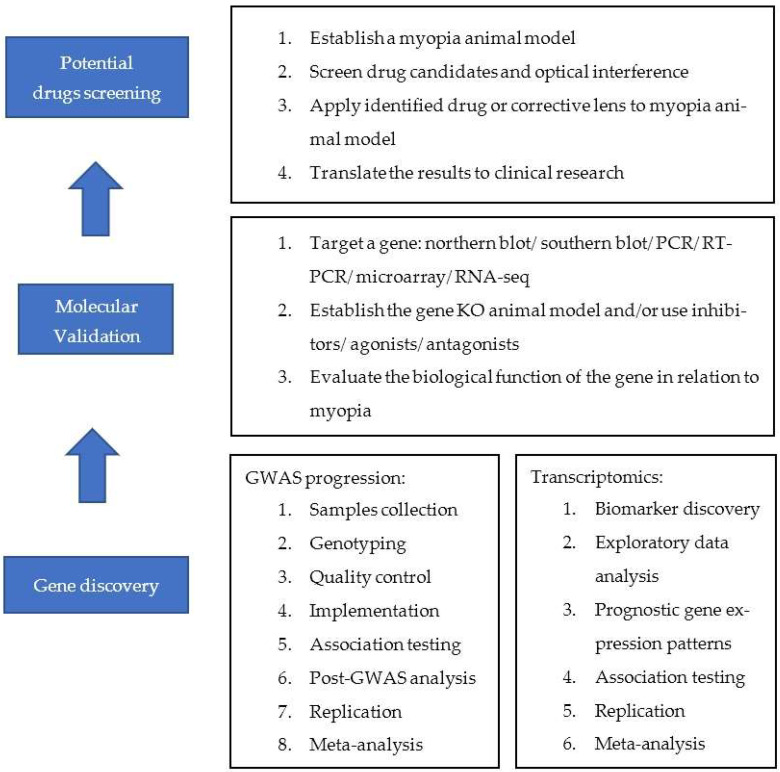
The general pipeline of precision medicine of a novel myopia therapy from gene discovery to clinical translation.

**Table 3 biomolecules-13-00494-t003:** Transcriptomics analysis in animal myopia model studies: from a few genes to signaling pathways identification.

Publications	Research Techniques	Animal Myopia Model, Targeted Tissues	Findings
PMID: 34405458 Karouta et al. (2021) [73]	RNA-seq; Enrichment analysis was undertaken using GO terms and the KEGG database; Validation of RNA-seq data by sqRT-PCR	Chick, FDM with and without drug interference, the retina	The *JAK-STAT* and circadian entrainment pathways showed significant gene enrichment through KEGG analysis; Alternative splicing events show commonality across growth-inhibitory treatments
PMID: 30167661 Srinivasalu et al. (2018) [74]	Customized microarray analysis; pathway analysis performed using GO; qPCR for validation	Guinea pig, FDM, forskolin (FSK) to induce myopia, sclera	FSK and FD both promote myopia through increased collagen degradation, targeting *cAMP* signaling pathway genes that could suppress myopia development.
PMID: 27832275 Metlapally et al. (2016) [75]	Custom multispecies miRNA microarrays and pathway analyses performed using GO, and qPCR for validation	C57BL/6J mice, FDM, the sclera	There were 54 differentially expressed miRNAs and 261 mRNAs in treated than in fellow eyes. Significant ontologies included intermediate filament organization, scaffold protein binding, detection of stimuli, calcium ion, G protein, and phototransduction. Significant differential expression of miRNAs Let-7a and miR-16-2, and genes *Smok4a*, *Prph2*, and *Gnat1* were confirmed.
PMID: 17652709 McGlinn et al. (2007) [76]	Affymetrix Chicken GeneChip microarray; qRT-PCR for validation.	Chick, FDM, the retina/RPE	In chickens that had 6 h of FDM *BMP2* and *NOV* downregulated 3 days of FDM, *BMP2*, *VIP*, *URP2*, and *MKP2* downregulated *ETB* and *IL-18* upregulated.
PMID: 17653032 Brand, Schaeffel and Feldkaemper (2007) [76]	Affymetrix GeneChip Mouse array, sqRT-PCR for validation	C57BL/6 mice, FDM while controlling for retinal illuminance	Affected genes at the mRNA level were *Egr-1*, *cFos*, *Akt2*, and *Mapk8ip3*. A pattern of differential transcription in the retina changed with the treatment.
PMID: 29625465 Srinivasalu et al. (2018) [77]	RNA-seq; Differential expression analysis using Voom; Pathway analysis using IPA; qRT-PCR for validation.	Guinea pig, FDM, peripapillary zone vs. peripheral temporal sclera	There were 348 genes differentially expressed between two regions, of which 61 were differentially expressed in the peripapillary zone between myopic and control eyes. Pathway analyses showed the involvement of Gαi signaling along with previously reported GABA and glutamate receptors.
PMID: 34830490 Zeng et al. (2021) [78]	ERG, RNA-seq, confocal microscopy, KEGG enrichment analysis	Guinea pigs, FDM, the retina	There were 288 genes upregulated and 119 genes downregulated in FDM retinas compared to naïve control. Tyrosine metabolism, ABC transporters, and inflammatory pathways were upregulated, whereas tight junction, lipid, and glycosaminoglycan biosynthesis were downregulated in FDM eyes.
PMID: 33674625 Vocale et al. (2021) [79]	RNA-seq; Enrichment analysis using GO, KEGG, Pathway Interaction Database, Reactome, and the Signal Transduction Knowledge Environment; microarray for validation	Chicks, FDM, the retina/RPE/choroid	FDM led to significant suppression in the ligand-gated chloride ion channel transport pathway via suppression of glycine, GABA-A, and GABA-C ionotropic receptors. Recovery from FDM for 6 h and 24 h all induced significant upregulation of cone receptor phototransduction, mitochondrial energy, and complement pathways.
PMID: 34841657 Liu et al. (2022) [80]	Differentially expressed miRNA (DE- miR) screening; Pathway analysis using GO, KEGG, UniProt, and Drugbank Protein-protein interaction (PPI) network creation and modular analysis; qRT-PCR	Mouse, FDM, Retina	Three upregulated miRNAs (mmu-miR-1936, mmu-miR-338-5p, and mmu-miR-673-3p) were significantly associated with myopia. GO functional analysis suggested these three miRNAs were targeted in genes mostly enriched in morphogenesis and developmental growth of retinal tissues.
PMID: 29987045 Wu et al. (2018) [81]	scRNA-Seq; RT-PCR; WB; GSEA	Mouse and guinea pig, FDM, sclera	The *eIF2*-signaling and *mTOR*-signaling pathways were activated in murine myopic sclera. Consistent with the role of hypoxic pathways in the mouse model of myopia, nearly one-third of human myopia risk genes from the GWAS and linkage analyses interact with genes in the *HIF-1α*–signaling pathway.

**Table 4 biomolecules-13-00494-t004:** Targeted genes in myopia animal models.

Publications	Research Techniques	Species Targeted Tissues	Findings
PMID: 15525903 Yin, Gentle, and McBrien (2004) [62]	PCR; Southern blot; Northern blot	Chick	No evidence of M1 mRNA expression in PCR or Southern blot. Chicks do not possess an M1 receptor, therefore pirenzepine, an M1 selective muscarinic antagonist is unlikely to exert its effect via the M1 pathway.
PMID: 32495406 Ding et al. (2020) [82]	DNA methylation assay; qPCR; Mass spectrometry	guinea pig, FDM, sclera	The methylation of four cytosine-guanine sites in the *IGF-1* gene promoter was significantly lower in the sclera after four weeks of FDM, and the transcription level of scleral *IGF-1* was moderately higher. The level of *MMP-2* mRNA in the sclera of MDT eyes was significantly higher, but not regulated by the methylation pathway.
PMID: 34083742 Quint et al. (2021) [83]	Eccentric photorefraction; scRNA-seq;	*gjd2a* and *gjd2b* mutant Zebrafish, the retina	Cx35.5 (*gjd2a*) depletion leads to hyperopia and electrophysiological changes in the retina and lack of Cx35.1 (*gjd2b*) led to a nuclear cataract that triggered axial elongation.
PMID: 19011237 Jobling et al. (2009) [84]	Collagen Gel cell contraction assay; qRT-PCR; Immunocytochemistry; WB	Tree shrew, FDM, sclera	*α-SMA* levels were increased in FDM eyes, suggesting increased numbers of contractile myofibroblasts, and decreased in eyes recovering from FDM.
PMID: 32725213 Srinivasalu et al. (2020) [85]	qPCR; cAMP radioimmunoassay	Guinea pig, FDM + drug interference, Sclera	*EP2* agonism increased *cAMP* and *HIF-1α* signaling, which decreased scleral fibrosis and promoted myopia. *EP2* antagonism instead inhibited these responses
PMID: 34287272 Hsu et al. (2021) [86]	ELISA; IHC and IF; WB	Golden Syrian hamsters, FDM, Retina	Resveratrol increased collagen I levels and suppresses the levels of *MMP2* and inflammatory cytokines such as *TNF-α*, *IL-6*, and *IL-1β*.
PMID: 33502461 Landis et al. (2021) [87]	HPLC; ddPCR; WB	C57BL/6J Mice, -10 D LIMunder 3 different lighting conditions, Retina	Lighting conditions affected mRNA levels of *TH* in both LIM and the control group of mice.
PMID: 29398596 Wei et al. (2018) [88]	ELISA; IF; IHC	Rat, chicken ovalbumin injection, Retina	The expression levels of inflammatory markers were up-regulated, lower refractive error and longer axial length were observed in eyes with allergic conjunctivitis, with myopia progression enhanced by *TNF-α* or *IL-6* administration.
PMID: 30029249 Yuan et al. (2018) [89]	qRT-PCR; WB; mechanical stretching protocol; confocal microscopy; Flow cytometry	Guinea Pig, FDM, Sclera	mRNA and protein levels of *RhoA* and *α-SMA* were significantly increased in the FDM eyes. Mechanical strain led to the activation of *RhoA* signaling and the differentiation of scleral myofibroblasts, likely to be mediated by *RhoA*/*ROCK2*-*MRTF-A*/ *SRF* pathway.
PMID: 21403852 Barathi and Beuerman (2011) [90]	qPCR; Northern blots	Balb/CJ (BJ) and C57BL/6 (B6) mice, LIM with and without atropine application, sclera	Muscarinic receptor subtype (M)1, 3, and 4 were upregulated in myopic sclera after atropine treatment.
PMID: 34608867 Summers and Martinez (2021) [91]	Microarray; qRT-PCR; ELISA	Chick, FDM with and without atropine application, choroid	*IL-6* was upregulated in chick choroids under recovery from induced myopia and compensation for positive lenses. Intraocular administration of atropine, an agent known to slow ocular elongation, also resulted in an increase in choroidal *IL-6* gene expression
PMID: 29163988 Li et al. (2017) [92]	qPCR; WB	Guinea pig, FDM with and without MT3 application, retina and choroid	MT3 can inhibit FDM, and MT3 treatment can result in changes in retinal and choroidal *TGF-β2* and *HAS2* mRNA and protein expressions.
PMID: 10644425 Bitzer, Feldkaemper and Schaeffel (2000) [93]	Northern blot; PCR	Chick, LIM and FDM, retina and choroid	*AHD2* is up-regulated after LIM. An inhibitor of RA synthesis, disulfiram, inhibited FDM but not LIM.
PMID: 33318135 Zhao et al. (2022) [43]	qRT-PCR; scRNA-seq; Pathway analysis using KEGG	*Vipr2*-KO mice, wild type C57BL/6J mice with and without *VIPR2* antagonist PG99-465 or *VIPR2* agonist Ro25-1553 treatment, FDM,	After either 1 or 2 days of FDM, retinal *VIP* mRNA expression was downregulated. scRNA-seq showed cAMP signaling pathway axis was inhibited in *VIPR2*-expressing cells. The selective *VIPR2* antagonist PG99-465 induced relative myopia, whereas the selective *VIPR2* agonist Ro25-1553 inhibited it. *Vipr2*-KO mice were myopic.
PMID: 35675393 Liu et al. (2022) [94]	Specific ablation and chemogenetic activation of intrinsically photosensitive retinal ganglion cells (ipRGCs); IHC; qRT-PCR; WB; ERG; HPLC	C57BL/6, *Opn4^Cre/Cre^*, *Opn4^Cre/+^*, *Opn4*^−/−^ mice, FDM, retina	FDM was inhibited in ipRGC-ablated and melanopsin-deficient animals. Melanopsin expression/photoresponses increased in form-deprived eyes. Cell subtype–specific ablation showed that M1 subtype cells, and probably M2/M3 subtype cells, are involved in ocular development.
PMID: 23550266 Tian et al. (2013) [95]	RT-PCR, WB	Guinea pig, LIM with or without *bFGF* injection, sclera	*bFGF* inhibited LIM. LIM decreased type I collagen, α2 integrin, and β1 integrin expressions while *bFGF* increased them significantly

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
