# Peer review of "Applications of Genomics and Transcriptomics in Precision Medicine for Myopia Control or Prevention"

_biomolecules, 2023, doi:10.3390/biom13030494_

Round 1

Reviewer 1 Report

This manuscript presents a very extensive catalog of research on genomics and transcriptomics of myopia in humans and animal models.  This manuscript may be a good resource for someone wanting to look for articles about different approaches to understanding myopia pathogenesis along with brief summaries of the different lines of research.  The abstract says that the strengths and weaknesses of the different approaches are considered, however, critical analysis of the relative value of the different lines of research is somewhat lacking. The authors only say that findings from some animal models may not be readily translatable to humans giving the example of the chick which does not possess a M1 receptor.  Further discussion of the limitations of various animal models would help to meet the expectations set up in the abstract. The abstract also says the potential applicability to precision medicine for myopia treatment are discussed.  However, this is discussed only in the most general terms that don’t really relate to specific research covered in the manuscript. 

The goal of genetic research, as outlined in this manuscript, is to discover the genes involved in myopia, evaluate the biological function of the genes and then to use that knowledge to develop interventions.  However, the authors of this manuscript do not mention an important recent success story in which genetic discoveries have led to a highly successful myopia intervention.  Rappon et al. (BJO 2022. doi: 10.1136/bjo-2021-321005) reported that the discovery of mutations in the cone opsin gene array led to the development Diffusion optics technology (DOT) lenses which were shown to slow myopia progression by 74% in a multicentre, randomised, controlled, double-masked trial. In that case, the biological function of the mutant genes was characterized (Greenwald et al. TVST 2017 Vol.6, 2. doi:https://doi.org/10.1167/tvst.6.3.2).  Moreover, mutations in the cone opsin genes were shown to account for about 4.6% of the variance in common myopia (Neitz et al. Genes. 2022; 13(6):942. https://doi.org/10.3390/genes13060942) which is 1.5 times more than the 2.9% identified by Kiefer et al. from the 23andme database which is cited in the manuscript.  This manuscript focuses on using genetic studies to find pharmacological interventions, however, effective optical interventions, informed by genetics, are already on the market.  These may go along way toward solving the myopia problem.

Reviewer 2 Report

This is an interesting review summarizing recent genomic and transcriptomic data important for myopia control or prevention. It correctly describes techniques and results used to advance this research field. To improve the manuscript the following points are important to be mentioned:

1. Syndromic and non-syndromic myopia should be better defined in the introduction. It should be mentioned that it is important to clinically characterize myopia patients to define if they have a syndromic or non-syndromic form of myopia.

2. The term myopia should be defined, how it is clinically diagnosed and more specifically, what is high and moderate myopia. A recent paper may help : https://www.sciencedirect.com/science/article/pii/S135094622200115X

3. The reader should understand why in animals FDM or LIM need to be performed to study myopia. A recent paper may help : https://www.mdpi.com/1422-0067/24/1/219.

4. The authors should be careful with electroretinogram (ERG) findings and association with myopia. It is a matter of debate if abnormalities in the ERG are also altered in myopia of if they only present alterations in the function of the retina. 

5. All tables need to be checked: gene should be in italics. The writing style differes: see e.g. page 9, 10, 11, 12, 13, 14. This should be homogenized.

6.please find below other important recent articles to be cited (and if possible to discussed):
(1) The epidemics of myopia: Aetiology and prevention.Review. Prog Retin Eye Res. 2018 Jan;62:134-149. doi: 10.1016/j.preteyeres.2017.09.004. Epub 2017 Sep 23.
Ian G Morgan, Amanda N French, Regan S Ashby, Xinxing Guo, Xiaohu Ding, Mingguang He, Kathryn A Rose
(2) Dopamine signaling and myopia development: what are the key challenges. Review. Prog. Retin. Eye Res., 61 (2017), pp. 60-71
X. Zhou, M.T. Pardue, P.M. Iuvone, J. Qu
(3) Myopia Genetics and Heredity. Review Children (Basel). 2022 Mar 9;9(3):382. doi: 10.3390/children9030382.
Yu-Meng Wang 1, Shi-Yao Lu 1 2, Xiu-Juan Zhang 1, Li-Jia Chen 1, Chi-Pui Pang 1, Jason C Yam 1 3

Reviewer 3 Report

Very interesting article by Jiang et al reviewing current data about the genomics and transcriptomics of myopia in humans and animal models, taking into consideration potential applicability for myopia treatment. However I do think that the authors should revise the article in terms of structure, as well as critically revise the Tables and add some Figures. I have explained my comments in more detail below.

Techniques of genomics and transcriptomics.
The order and structure of this part is a bit weird. I would start with the significance of genetic analyses before moving on to the different techniques used. 
For the genomics section, I would suggest to begin with linkage analysis followed by GWAS (since this was how it chronologically done). Then I would also add microarray (which is now in the transcriptomics section) to this section.

Applications of new technologies to myopia research.
Only the big GWAS studies are mentioned. Add references on the first GWASs in Europeans and Asians here? Also, a big GWAS meta-analysis PMID by the UK Biobank (PMID 32231278) is being left out and should be added. 

Table 1/2/3
These tables need to be revised and are not acceptable in its current form. There's too much text in the tables which doesnt make it very clear.

I would suggest to use first authors and year of publication, along with a PMID and cohort/study size instead of the whole title of the paper. Also research techniques should be shorted (especially in Table 1). Please think carefully about the information you want to exhibit here. Also I am not sure about the completeness of table 1. The alignment of these tables also gives room for improvement.

I would like to challenge the authors to come up with a few nice figures for this paper to summarize the findings (instead of a table). I think the paper would really benefit from this. 

Round 2

Reviewer 1 Report

All my concerns we remedied in the revised manuscript.

Reviewer 2 Report

The authors made the changes as requested.

Reviewer 3 Report

I thank the authors for changing the paper according to my suggestions. The tables look a bit better, although I am still not too happy with them (espec table 3 contains too much text). I like the figure that was adde by the authors. One minor thing, the references have a different layout/font.